# Incorporation of *Prosopis cineraria* Extract Improved the Mechanical, Barrier and Antioxidant Properties but Not the Antibacterial Activity of *Tigertooth croaker* Fish Scale Gelatin Film [note 1]

**DOI:** 10.3390/foods13040538

**Published:** 2024-02-09

**Authors:** Zeinab Kord, Ali Taheri, Mostafa Ghaffari, Salim Sharifian

**Affiliations:** Fish Processing Technology Division, Faculty of Marine Sciences, Chabahar Maritime University, Chabahar 99717-78631, Iran; kord.zeynab97@gmail.com (Z.K.); mgmostafaghaffari@gmail.com (M.G.); sharifian.s@cmu.ac.ir (S.S.)

**Keywords:** croaker fish, scale gelatin, Prosopis extract, edible film, mechanical properties, antioxidant

## Abstract

Scale gelatin films derived from croaker fish and infused with *Prosopis* (*Prosopis cineraria*) extract (PE) at concentrations of 0.3% and 0.7% were produced. A control film, void of extract, was employed for comparative purposes. The thickness of each film was found to be statistically insignificant (*p* > 0.05). The results show that the highest solubility (78.57 ± 3.57%) was found for the glycerol film, and the least permeability was found for the water vapor (0.74 ± 0.09 ×10^−10^g s^−1^m^−1^Pa^−1^); however, the water vapor permeability (WVP) and water solubility (WS) of the films that contained PE were considerably lower than those of the control film (*p* < 0.05). In contrast to the control film, those infused with 0.7% PE exhibited exceptional UV-barrier properties (>99%) and favorable thermal characteristics. The highest and lowest antioxidant activities were found for the 7% *Prosopis cineraria* extract (56.96 ± 2.6%) and the glycerol film (40.66 ± 2.46%), respectively. No antibacterial activity was observed in these films. Microscopic pictures showed that all three films had a uniform and plain surface. Fourier transform infrared spectroscopy–attenuated total reflectance (FTIR–ATR) analysis revealed distinct amide bands and protein–polyphenol interactions within the films that contained the extract.

## 1. Introduction

As the main material for the food packaging industry, plastics have caused many concerns about environmental pollution and related ecological problems worldwide. Non-biodegradable materials are not subject to decomposition by living organisms and have persisted in the environment for numerous years. In recent years, due to heightened public awareness and the implementation of environmental regulations, degradable materials have emerged as a secure and beneficial alternative to synthetic packaging materials derived from petrochemical products [1]. Different innovative biodegradable materials, including polysaccharides, lipids, and proteins or their mixture have been examined to develop packaging and edible films used in the food industry [2]. Among the various natural biopolymers, gelatin, as a proteinaceous biopolymer of natural origin, has recently garnered increased attention, especially as a material for film formation, owing to its remarkable film-forming ability and outstanding gas barrier [3]. Gelatin is typically derived from the skin, bone, and connective tissue of swine and bovine. However, given the limited application of land animal-derived gelatin in certain religions, the extraction of gelatin from marine organisms, particularly fish, is being recognized as an essential and untapped resource for the production of edible films [4]. In the last decade, a variety of global initiatives have been undertaken to transform the by-products of fish processing into value-added products, including but not limited to edible gelatin films [3,5]. Prior research has demonstrated that producing gelatin from fish by-products, such as scales, can enhance the value of these by-products while mitigating the environmental challenges associated with marine waste [6,7]. However, the film derived from fish gelatin exhibited functional properties that were comparatively weaker, particularly with regard to water resistance, in contrast to those of mammalian origin [5]. In recent years, several natural cross-linking agents, including phenolic compounds [8], seaweed extract [4] and plant-based essential oils [1,2], have been utilized to enhance the mechanical and functional properties of gelatin films from marine sources. By incorporating bioactive compounds into gelatin film, it would be possible to confer antioxidant and antimicrobial activities, thereby extending the shelf-life of food products, in addition to generating good cross-linking effects [9].

Herbal medicines are the best known and well-documented natural antioxidant sources. They are rich in secondary metabolites such as flavonoids and other phenolic compounds, which can act as antioxidants and extend the shelf lives of food products. Among different medicinal plants, *Prosopis* plants have been used in the food and pharmaceutical industries. *Prosopis cineraria*, known as Ghaf in Arabic and Kahour in Iran, is a species of *Prosopis* genus and native to some countries such as Pakistan, United Arab Emirates, and Iran. Numerous and valuable properties have been mentioned for this plant in folk medicine [10]. Furthermore, it was reported that the leaves and seeds of these plants are a rich source of phytochemicals, namely flavonoids, tannins, alkaloids, quinones, and phenolics [11].

Previous studies have demonstrated that incorporating some plant extract such as rosemary [12] and green tea [13] in fish gelatin films could enhance the film’s mechanical and antioxidant properties. Most of these plant extracts are rich in secondary metabolites, especially the phenolic compounds that could play an important role in improving film properties [7]. However, the practical use of fish gelatin incorporated with plant extracts needs to be further investigated. The use of *Prosopis* leaves extract, as a good source of natural antioxidants, may improve physical properties and antioxidant ability of the gelatin film. 

Also, Tigertooth croaker fish (*Otolithes ruber*; *Sciaenidae*) is distributed in the Oman Sea, the Persian Gulf, Indian, and Pacific oceans as a demersal species [14]. The catch data of *Otolithes ruber* from four Iranian coastal regions of Khuzestan, Hormozgan, Bushehr, and Sistan-Baluchestan are 8530 tons [15]. This is an economic fish used for filleting in the fish processing factories of Iran and the detached scales in the filleting process could be used to produce value-added products. It is well known that fish scales are a rich source of type I collagen, which can be used as a potential source of gelatin for edible film making. So, the objective of this study was to investigate the effect of *Prosopis* leaf extract on the physical, antioxidant and antimicrobial properties of film produced with gelatin extracted from *Otolithes ruber* scales. 

## 2. Materials and Methods

### 2.1. Chemicals

2,2-diphenyl-1-picryl hydrazyl (DPPH) solution, NaOH, acetic acid, NaBr, KBr, potassium ferricyanide and trichloroacetic acid were purchased Sigma-Aldrich (St. Louis, MO, USA). 

### 2.2. Fish Scale Preparation

The scales of croaker fish (*Otolithes ruber*) were taken from a fish processing factory (Protein Arman Jonoub Co., Ltd., Tehran, Iran) located in Chabahar, Sistan and Balouchestan Province, Iran. The scales were collected after filleting and brought to the biotechnology laboratory of Chabahar Maritime University on ice within 40 min. Upon arrival, they were washed with chilled tap water, packed in zip keep bags, and stored in a freezer at –20 ± 1 °C for further study. 

### 2.3. Preparation of Prosopis Extract

Mesquite (*Prosopis cineraria*) leaves were collected from the coastal regions of Chabahar, Iran. After washing with tap water, the leaves were dried in the shade and pulverized to a fine powder. *Prosopis* extract (PE) was prepared according to Santoso et al. [16] method with some modifications. Briefly, 10 g of the leaf powder was added to a mixture of ethanol 96% and water (*v*/*v* 1:1) in a ratio of 1:10 powder/solvent (*w*/*v*), and stirred at 20 °C for 48 h. The mixture was filtrated with Whatman filter paper No.1, dried overnight in a laminar flow hood (Boyikang, Beijing, China) and stored at −20 °C for more analysis. Before incorporation, the crude extracts were weighed, dissolved in distilled water, and centrifuged at 5000× *g* (4 °C, 10 min) to remove impurities. 

### 2.4. Gelatin Extraction Procedure

The scales of croaker fish were used for gelatin extraction according to Thuy et al. [17] method with some slight modifications. Briefly, scales were soaked in 0.1 M NaOH at a ratio of 1:3 (*w*/*v*) and placed at 4 °C for 8 h to cast off the non-collagenous protein and excess tissue once they had been swollen. The scales were subjected to an alkalinization process, followed by washing with neutral pH water. Demineralization was achieved by treating the scales with a 0.5 M acetic acid solution at a concentration of 1:3 *w*/*v*, for 48 h at 4 °C. The scales were then re-washed with tap water to attain a neutral pH. Swollen scales were immersed in hot distilled water at 90 °C for 6 h. The resulting solution was subjected to centrifugation at 15,000× *g* for 15 min at 4 °C to separate the solid scales from the liquid phase. The supernatant was then collected, freeze-dried using a Jalteb, Iran apparatus, and designated as gelatin powder.

### 2.5. Film Preparation

Four grams of the gelatin powder obtained in the previous step was dissolved in 100 mL distilled water (45 °C) to prepare the film-forming solution (FFS). The FFS was plasticized by adding glycerol (25% *w*/*v*) and stirring continuously at 45 °C for 30 min. The *Prosopis* extract (PE) was added to the FFS to obtain the final concentrations of 0, 0.3 and 0.7% (*w*/*v*). In order to mitigate the formation of air bubbles in the FFS, a centrifugal vacuum mixer was employed. Subsequently, the resulting mixture was cast onto a Plexiglas sheet and left to dry at ambient temperature. Prior to conducting mechanical property tests, the films were stored in desiccators that contained saturated NaBr solutions with a relative humidity of 58% for a duration of two days, again at room temperature. As a control, Gelatin film without PE was prepared using the same methodology.

### 2.6. Gelatin Films Characterization

#### 2.6.1. Film Thickness

The films’ thickness was measured at five randomly selected locations using a 7300 series micrometer (Mitutoyo Co., Kanagawa, Japan). 

#### 2.6.2. Water Solubility

Film solubility was measured according to the method of Yanwong and Threepopnatkul [18]. Nine pieces (20 × 20 mm) of the film were dried at 100 °C for 24 h and weighed (W1). Each dried film sample was immersed in 50 mL of distilled water for 24 h. The samples were then filtered using Whatman No. 1. To obtain the weight of the dry matter (W2), the filter paper containing insolubilized portions of film was dried in an oven (100 °C, 24 h), and finally, the solubilized dry matter (Ws) calculation was performed based on Equation (1): (1)$$\mathrm{Ws}\;(\%)=\frac{\left((W1-W2)\times 100\right)}{W1}$$

#### 2.6.3. Water Vapor Permeability (WVP) Measurement 

The WVP of different films was measured according to the method described by Wu et al. [13] with a slight modification. The gelatin film samples were first cut into a circle and then set on a glass permeability cell with an inner diameter of 5 cm and fastened using two silicone O-rings. Before mounting the film, 6 mL of distilled water was added to the cell to create a relative humidity close to saturation. An environment with 55% RH was prepared by setting silica gel in a desiccator. After placing the cell in the desiccator, the weight changes of the film were recorded for 10 h (every one hour) using an analytical balance. The water vapor permeability rate was calculated by dividing the slope of the weight gain vs. time plot. Three replicates were considered for each film, and WVP was calculated based on the Formula (2):(2)$$WVP=\frac{\left(W\times X\right)}{\left(A\times ∆P\right)}$$
W = the weight gain of the cup by the slope (g s^−1^);X = the film thickness (m);A = the area of exposed film (m^2^);∆P = the vapor pressure differential across the film (Pa).


#### 2.6.4. Opacity

Film opacity (O) was measured according to the method of Gómez-Estaca et al. [12]. Pieces of each film with sizes of 50 × 200 mm were prepared and placed in a test cell of a spectrophotometer. The absorbance was read at 600 nm using a UV 160A spectrophotometer (Shimadzu, Croissy, France) and the opacity was calculated based on the Formula (3): (3)$$O=\frac{Abs\;600}{X}$$
where Abs600 is the absorbance at 600 nm and X is the thickness of the film (mm). Measurements were carried out in triplicate. 

### 2.7. Microstructure Analysis by Scanning Electron Microscopy (SEM)

Longitudinal and transverse cross-sectional photography of film samples was performed using Hitachi S-4800 scanning electron microscope, as Weng and Zheng [19] described. The samples were dried in a desiccator, mounted on a bronze stub and covered with a layer of gold to create conductivity. The observation was carried out at an acceleration voltage of 15 kV. 

### 2.8. Structural Interactions by Fourier Transform Infrared Spectroscopy-Attenuated Total Reflectance (FTIR-ATR)

FTIR–ATR spectroscopy (Spectrum-RXI) was used to investigate the structural interactions of gelatin films containing PE. The scanning was performed using the spectrum-RXI FT-IR model in the range of 1400–400 cm^−1^ (resolution of 14 cm). The samples were prepared in powder form, KBr was added and the mixture was prepared as a tablet. The tablets were placed in the FTIR apparatus and the absorbance spectrum was read.

### 2.9. Thermal Properties 

The samples were powdered in liquid nitrogen and kept in a desiccator for 2–3 days. After that, the thermal properties of the films were measured using a NETZSCH-Maia 200-F3 (Selb, Germany) differential scanning calorimeter within the temperature range of 30–250 °C and a heating rate of 10 °C/min. 

### 2.10. Determination of Antioxidant Activity 

The present study investigated the antioxidant capabilities of the gelatin films through the utilization of two distinct methods of free radical scavenging, namely DPPH radical scavenging activity and reducing power. The sample solution was prepared by precisely measuring 1 mg of the gelatin films, dissolving them in 1 mL of distilled water and subsequently centrifuging the solution at 2700× *g* for a duration of 10 min. The supernatant thus obtained was employed for conducting the antioxidant tests. All the assays were carried out in triplicate to ensure the reliability and accuracy of the results.

DPPH free radical scavenging activity was measured as the method described by Tongnuanchan et al. [1] with a slight modification. The sample and DPPH solution (0.15 mM in 95% ethanol) in a ratio of 1:1 (*v*/*v*) were mixed and placed in the dark (room temperature, 30 min). After that, its absorbance was read at 517 nm (UV 160A, Shimadzu, Croissy, France) against 95% methanol as blank. The radical scavenging activity was calculated based on the Formula (4): (4)$$\mathrm{Radical}\;\mathrm{scavenging}\;\mathrm{activity}\;(\%)=\frac{\left(A0-A\right)}{A0}\times 100$$
where A0 is the absorbance of the control and A is the absorbance of the sample.

The reduction in power in the film samples was executed following the method outlined by Pires et al. [20]. In summary, 1 mL of the prepared sample was mixed with 1 mL of potassium ferricyanide (1%) and 1 mL of 0.2 mM phosphate buffer at pH 6.6. The mixture was then incubated at a temperature of 50 °C for a period of 20 min. Following incubation, 1 mL of 10% trichloroacetic acid (TCA) was introduced to the solution. Subsequently, 2 mL of distilled water was added to the 2 mL of the final solution, and 0.4 mL of FeCl_3_ solution (0.1%) was included and allowed to remain at room temperature for 10 min. The optical absorption of the samples at 700 nm was ultimately measured using a UV 160A spectrophotometer (Shimadzu, Croissy, France).

### 2.11. Determination of Antimicrobial Activity 

The antibacterial activity on films was measured using the disc diffusion method as described by Martucci et al. [21]. Experiments were performed with two Gram-negative bacterial strains, including *Staphylococcus aureus* (ATCC 6538) and *Listeria monocytogenes* (CECT 5873) and two Gram-positive bacterial strains, including *Pseudomonas aeruginosa* (ATCC 9027) and *Escherichia coli* (ATCC 8739). Bacterial strains procured in lyophilized form were obtained from the esteemed Pasture Institute in Iran. A volume of 100 mL of inoculum, containing 107 colony-forming units per milliliter of the tested bacteria, was uniformly dispersed onto tryptic soy agar (TSA) or plate count agar (PCA), and subsequently, sterile blank discs were fixed onto the surfaces of the plates. An amount of 15 µL of the extract solution was added onto the blank discs. Subsequent to this, the plates were incubated at a temperature of 37 °C for a period of 24–36 h. The diameter of the inhibition zone was measured utilizing a caliper. The experimental tests were conducted thrice to ensure the validity and reliability of the results.

### 2.12. Statistical Analysis

The data were reported as means with standard deviation. One-way ANOVA test was used to compare treatments and Duncan multiple range test was used to compare mean values (*p* < 0.05). All statistical tests were carried out by SPSS 18 (SPSS Inc., Chicago, IL, USA).

## 3. Results and Discussion

### 3.1. Thickness, Water Vapor Permeability (WVP) and Water Solubility (WS) 

The thickness measurements of various croaker scale gelatin films are presented in Table 1, with values ranging from 0.22 mm in the control group to 0.23 mm in the film supplemented with 0.7% PE. Upon conducting statistical analyses, it was found that the addition of PE did not elicit a statistically significant impact on the thickness of the films. This outcome aligns with previous research conducted by Wu et al. [13], who similarly observed no significant difference in the thicknesses of gelatin films enriched with tea extract relative to those composed of pure gelatin.

WVP is essential in understanding the humidity change between the film on the food product and the surrounding environment [22]. This property greatly impacts the food shelf-life [23]. The WVP of different films was measured at room temperature. Adding the PE significantly affected the WVP of films and decreased it from 0.78 to 0.74 × 10^−10^ g s^−1^ m^−1^ Pa^−1^ (Table 1). Reduction in WVP of film incorporated with 0.7% PE might be due to the presence of phenolic compounds abundant in the plant extract [11]. It can establish cross-links in the gelatin matrix through hydrogen bonding and hydrophobic interactions [4]. Similar results were reported for WVP changes in silver carp skin gelatin film incorporated with green tea extract [13] and fish skin gelatin film incorporated with seaweed extract [4].

Water solubility (WS) is a critical characteristic of biodegradable films, as it ascertains the extent of film resilience to water, particularly in settings that are moisture-laden [24]. The WS of gelatin films that were merged with PE (0.3 and 0.7%) are provided in Table 1. Normally, the WS of gelatin films that were combined with PE was inferior to the value of the control (*p* < 0.05) and decreased with the progressive concentration of *Prosopis* extract from 0.3 to 0.7% (*p* < 0.05). The reduction in the film’s dissolution in water is a reflection of the firmness of the film network. Protein–polyphenol interactions are one of the significant forces that contribute to network stability. It is plausible that the diminished water solubility in the film with 0.7% PE was owing to these interactions [24].

Similarly, Govindaswamy et al. [25] reported that the incorporation of brown seaweed fucoidan had significantly reduced the water solubility of carp swim bladder gelatin films from 91.49 to 82.05%. In a separate investigation, the integration of diverse herbaceous extracts such as clove, cinnamon, and star anise extracts into cuttlefish skin gelatin films resulted in a marked decline in water solubility [24]. It is well documented that the cross-linking of gelatin film with phenolic compounds, enzymes, and some other chemicals decreases the water solubility due to reduction in low molecular fractions. Furthermore, the extract’s phenolic compounds possess the ability to diminish solubility by enhancing the interactions among gelatin molecules via a covalent bond [25]. 

### 3.2. Optical Properties

The light transmittance of fish gelatin films which have integrated *Prosopis* extract are presented in Table 2, at wavelengths spanning from 200 to 800 nm, encompassing UV and visible light. Nevertheless, a dearth and insignificance of light transmission occurred at 200 nm and 280 nm for the films. These results are comparable with those of a study by Hoque et al. [24]. Govindaswamy et al. [25] also reported low UV light transmittance for fish gelatin films incorporated with seaweed fucoidan. Prior research has indicated that protein films containing high levels of aromatic amino acids possess exceptional UV deterrent properties, owing to their ability to absorb UV light [25,26]. While the transmission of UV light is prevented by this phenomenon, the transmission of light in the visible range of 350–800 nm is quite high for all gelatin films, ranging from 60.63% to 78.62%. The incorporation of *Prosopis* extract did not have a significant impact on the visible light transmission of the films, as evidenced by the lack of difference in transparency values between the various films (*p* > 0.05). This could potentially be attributed to the formation of a uniform network within the films, which in turn permits the easy passage of light through them [24].

### 3.3. Scanning Electron Microscopy (SEM)

SEM pictures of the cross-section and surface of croaker gelatin films incorporated with *Prosopis* extracts and the control are shown in Figure 1. In general, the surfaces of all films were smooth, relatively homogeneous, with no marked difference between the control film and those incorporated with *Prosopis* extracts. The cross-sectional analysis of gelatin films infused with *Prosopis* extract revealed denser structures in comparison to the control film. Moreover, the structural heterogeneity observed in the 0.7% PE film exceeded that of the film infused with 0.3% *Prosopis* extract. This phenomenon of compact structure formation in the film could be attributed to the heightened occurrence of protein–polyphenol interactions [24]. Similarly, Rattaya et al. [4] reported a more compact cross-sectional structure, which works on fish gelatin film incorporated with seaweed extract. These researchers postulated that the observed phenomenon may be attributed to the augmented bonding between phenolic compounds and protein strands through covalent and non-covalent interactions.

### 3.4. Fourier Transforms Infrared Spectroscopy (FTIR) 

Fourier transform infrared (FTIR) spectroscopy is a non-invasive and expeditious methodology that furnishes significant insights into the functional groups of film samples. Furthermore, the FTIR data have the capacity to investigate alterations in the structures of these films [27]. FTIR spectra of gelatin films incorporated with *Prosopis* extracts and the control are shown in Figure 2. Amides A and B and I–III are five characteristic infrared (IR) absorption bands that are usually reported in gelatin conformational studies. The wavenumbers of the amide I, II and III bands are related to collagen configuration [28,29]. In the present study, amide-I bands representing C=O stretching vibration coupled with the CN stretch appeared at the wavenumbers of 1654.86, 1670.82 and 1664.79 cm^−1^ for the control, 0.3% PE and 0.7% PE films, respectively. The amide I band in PE incorporated films shifted to the higher wavenumber.

Similarly, Rasid et al. [30] reported an increase for the amide I wavenumbers of gelatin films incorporated with *Centella asiatica* (L.) urban extract and concluded it might be due to an increase in C=C stretching within the aromatic ring, which indicates the functional group of phenolic compounds of the extract. Amide II band wavenumbers emanated from N–H bending and C–N stretching vibrations [31]. These band for the control film, 0.3% PE and 0.7% PE films were observed at 1541.95, 1536.77 and 1543.64 cm^−1^, respectively. The shifting of amide groups within the herb extract integrated films recommended interactions between a number of the N–H groups of gelatin protein and hydroxyl groups of phenolic compounds to form hydrogen bonds [32]. The amide III bands, which appeared at a wavenumber of around 1236 cm^−1^ in the fish collagen films, can referred to vibrations in C–N and N–H groups of amide bands [24]. As shown in Figure 2, the amide A bands at peaks of 3406.29, 3426.31 and 3414.21 appeared for the control, 0.3% PE and 0.7% PE incorporated films, respectively. The shift of amide A bands in the 0.7% HE incorporated film to a lower wavenumber might be due to changes in the secondary protein structure. It is well known that a plant extract encourages the interaction between phenolic compounds and NH_2_ and OH groups in gelatin, and leads to crosslinking in the gelatin network [24,30]. Furthermore, amide B was also observed in the spectra at 3275 cm^−1^, representing the CH and -NH_2_ functional groups [24]. Our result is comparable with the study by Muyonga et al. [33], who observed amide A and amide B bands in adult Nile perch collagen at the wavenumbers of 3458 and 2926, respectively. Finally, the wavelength range of 2100–2260 in the control film and the film containing 0.7% PE represents variable alkynes [34]. 

The augmentation in the amide A, amide I, amide II, and amide III bands in fish gelatin films that were infused with *Prosopis* extract could potentially be related to the gelatin–phenolic compound interactions of the extract. The Fourier transform infrared (FTIR) analysis exhibited that the inclusion of the extract led to the formation of hydrogen bonds with diverse functional groups which consequently bolstered the intermolecular connections in the gelatin films. This resulted in an enhancement of the mechanical and water resistance properties of the films. 

### 3.5. Antioxidant Properties

The results of the DPPH radical scavenging activity and the reducing power of the films are shown in Figure 3A,B. As shown in Figure 3A, a significant DPPH activity (40.66%) was observed in the control film (without any extract), which was in agreement with the study of Hanani et al. [23], who reported a radical scavenging activity of 53% for fish gelatin films. The antioxidant properties exhibited by marine fish proteins are attributed to the presence of particular peptides including glycine and proline amino acids [35]. The films that contained 0.3% and 0.7% PE exhibited significantly higher scavenging activities, specifically at 51.7% and 58.96%, respectively, when compared to the control film. Additionally, an increase in extract concentration resulted in a significant increase in the DPPH scavenging activity of the films (*p* < 0.05). A study conducted by Rasid et al. [30] revealed that incorporating *Centella asiatica* (L.) urban extract significantly enhanced the DPPH scavenging activity of fish gelatin films in comparison with the control sample. The antioxidant properties of the films with the extract might be attributed to interactions between phenolic compounds and gelatin’s amino acids. The well-established antioxidant capacity of phenolic compounds can be attributed to their characteristics as reducing agents, hydrogen donors, metal chelators and singlet oxygen quenchers [23,32]. 

The ability of fish scale gelatin films with *Prosopis* extract and the control to reduce ferric ion (Fe^3+^) is shown in Figure 3B. The reduction power in the control gelatin film (without extract) was low and negligible (0.01), while the incorporation of 0.3 and 0.7% *Prosopis* extracts significantly increased the reduction power to 0.18 and 1.20, respectively. Furthermore, the reduction power increased by increasing the concentrations of film extracts (*p* < 0.05). Mohan et al. [36] reported a significantly high reduction power in *Prosopis* leaf extract, which may be attributed to the phenolic content of the extracts. 

The gelatin films containing 0.3, 0.7% PE and control film showed no antimicrobial activity against the *Escherichia coli*, *Listeria monocytogenes* and *Pseudomonas aeruginosa* strains studied. Contrary to our results, previous studies showed that the *Prosopis* extract had high activity against different strains of bacteria [34,37,38]. On the other hand, the FTIR analysis showed that there were metabolites with antimicrobial activity such as phenolic compounds and other OH-containing compounds in the films. The absence of microbial activity could plausibly be attributed to the insufficient concentration of the extract that was integrated into the films, thereby rendering them devoid of antimicrobial properties. In keeping with our findings, Núñez-Flores et al. [39] documented a lack of association between the antioxidant and bactericidal activity of fish gelatin films that contained lignin.

### 3.6. Thermal Properties

DSC thermographs, glass transition temperature (Tg), melting transition temperature (T_max_) and the enthalpy of fish gelatin films incorporated with 0.3 and 0.7% *Prosopis* extract (PE) are shown in Figure 4 and Table 3. As shown in Figure 4, all film samples represented a Tg and an endothermic melting transition (T_max_). The determination of Tg is of the utmost importance in comprehending the characteristics of a polymer. It is unambiguously linked to the molecular segmental motion of the amorphous structure of the films [2]. The Tg value of the control film was observed at a temperature of 67.13 °C which was higher than the commercial tilapia scale gelatin (64.3 °C; [32]) and Baramundi (*Lates calcarifer*) scale gelatin (59.23 °C; [40]), but lower than the tilapia scale (137 °C; [41]) and skin gelatins (71.27 °C; [5]). A significant lower Tg (59.78 °C) was found in film with 0.3% PE compared to the control film and film with 0.7% PE (67 °C). The decrease Tg in the film with 0.3% PE is probably because of water, which acts as a plasticizer and reduces the glass transition temperature by increasing the space among gelatin polymer chains [42]. Furthermore, T_max_ and enthalpy of films decreased from 71.83 to 63.47 °C as the concentration of extract incorporated in the films decreased from 0.7 to 0.3%. The better T_max_ and ∆H observed within the films are probably because of extra inter-chain interaction of gelatin stands, usually through a hydrophobic interaction and hydrogen bond. The result was in agreement with Rasid et al. [30] who found that *Centella asiatica* (L.) urban extract possesses the thermal stability of gelatin-based film through protein–phenol interaction. 

### 3.7. Antimicrobial Properties

In this particular investigation, it was found that the gelatin films incorporating 0.3% and 0.7% of *Prosopis cineraria* extract did not exhibit any antibacterial activity against *Listeria monocytogenes*, *E. coli* and *Pseudomonas aeruginosa*. Saadoun et al. [43] undertook a study whereby they employed methanol solvent to extract fresh and dry *Prosopis juliflora* plant and tested it against various strains of pathogenic bacteria. The maximum amount of non-growth halo was noted against Gram-positive bacteria in comparison to Gram-negative bacteria. The highest halo of non-growth associated with the fresh extract of mesquite plant was observed in *Streptococcus* and *Bacillus*, with halos of 22 and 19 mm, respectively. The lack of antibacterial activity in this study may be attributed to the quantity of extracts used and the choice of solvent. Wu et al. [13] also utilized green tea extract in the same gelatin films; however, no antibacterial activity was detected against the tested bacteria.

## 4. Conclusions

Until the present time, no scholarly investigation had been conducted on the supplementation of *Prosopis* extract to enhance the properties of the fish gelatin film. According to the findings, the inclusion of *Prosopis* extract had a positive impact on the physio-mechanical characteristics of the gelatin films derived from the croaker fish scales. The film that was supplemented with PE demonstrated exceptional barrier properties against moisture and UV light. The concentration of extract at 0.3% exhibited a superior thermal stability in the film. Moreover, the addition of PE enriched the composite film with antioxidant activity, although the quantity of PE (0.3% and 0.7%) was insufficient to demonstrate antibacterial activity. Thus, it can be inferred that *Prosopis* extracts can serve as a natural additive to enhance the characteristics of gelatin film obtained from croaker fish scales. Future research endeavors could be carried out to broaden the scope of industrial recommendations by studying the application of the incorporated gelatin film in fish fillet packaging.

## Figures and Tables

**Figure 1 foods-13-00538-f001:**
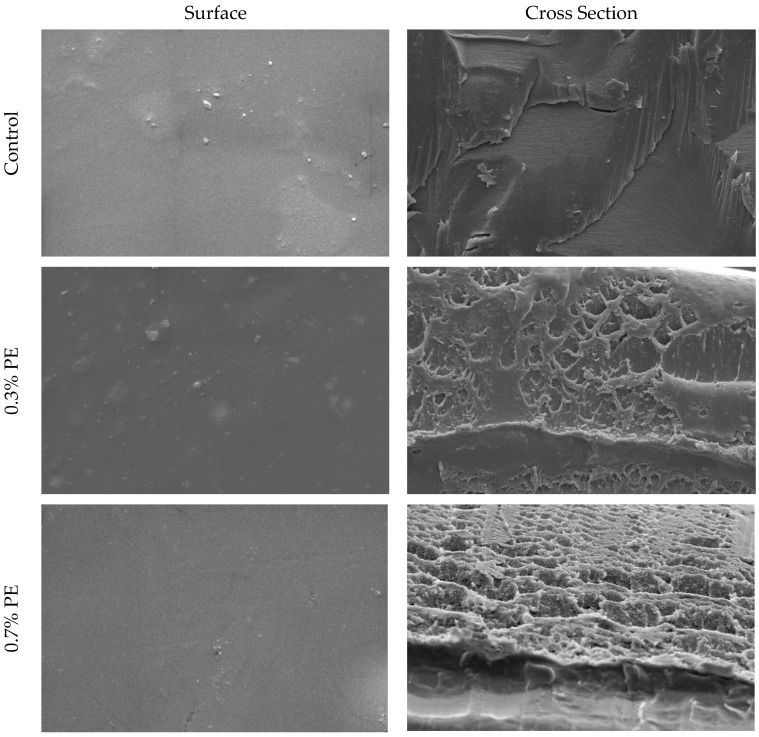
SEM micrographs of the surface and cross-section of croaker fish scale gelatin films incorporated with *Prosopis* extracts (PE) and the control.

**Figure 2 foods-13-00538-f002:**
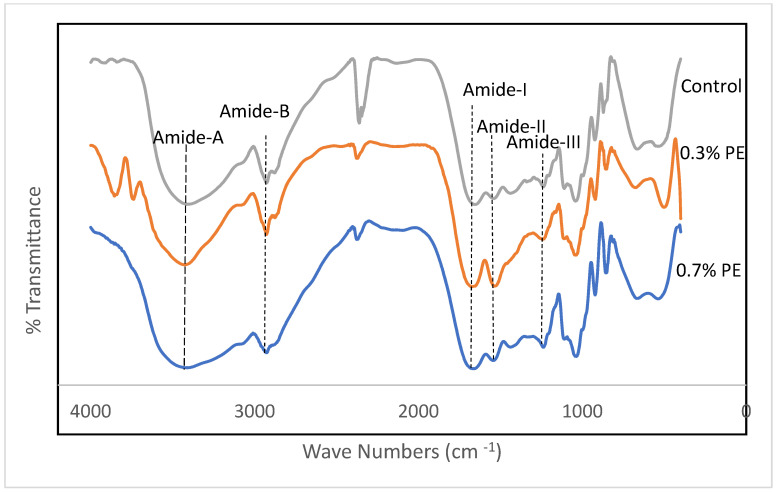
FTIR spectra of croaker fish scale gelatin films incorporated with *Prosopis* extracts (PE).

**Figure 3 foods-13-00538-f003:**
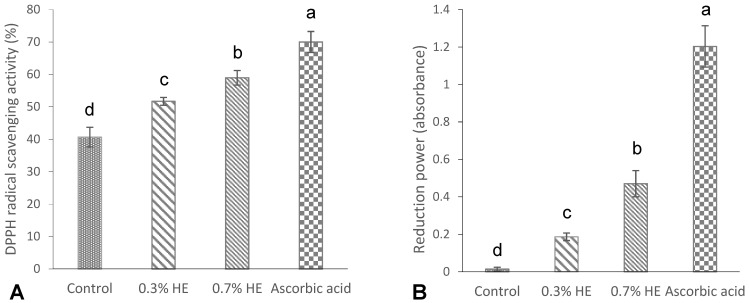
DPPH radical scavenging activity (%) (**A**), and reducing power (absorbance) (**B**) of croaker fish scale gelatin films incorporated with 0.3 and 0.7% *Prosopis* extracts (PEs) and the control. Different letters of a–d, indicate a significance difference at *p* < 0.05.

**Figure 4 foods-13-00538-f004:**
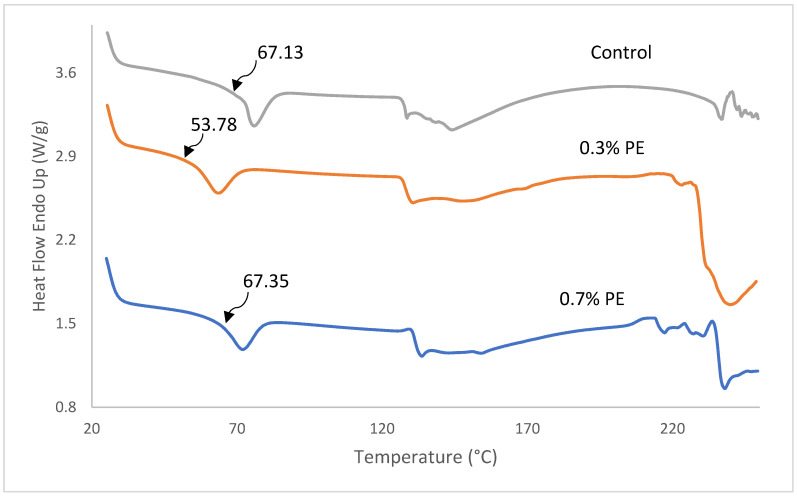
DSC thermographs of fish gelatin films incorporated with *Prosopis* extracts and the control. Tg values are shown by arrows.

**Table 1 foods-13-00538-t001:** Thickness, water vapor permeability (WVP) and water solubility (WS) of croaker scale gelatin films incorporated with *Prosopis* extract (PE).

Film Type	Thickness (mm)	WVP (×10^−10^ g s^−1^ m^−1^ Pa^−1^)	WS (%)
Control	0.22 ± 0.01 ^a^	0.78± 0.02 ^a^	78.57 ± 1.98 ^a^
0.3% PE	0.22 ± 0.01 ^a^	0.77 ± 0.04 ^ab^	71.33 ± 2.31 ^b^
0.7% PE	0.23 ± 0.01 ^a^	0.74 ± 0.01 ^b^	65.28 ± 2.81 ^c^

Values are expressed as mean ± standard deviation. Different letters in the same column indicate a significance difference at *p* < 0.05.

**Table 2 foods-13-00538-t002:** Light transmittance (%) and transparency values of fish gelatin films incorporated with herb extracts.

Film Type	Wavelength (nm)								
	200	280	350	400	500	600	700	800	Transparency Values *
Control	1.02	5.57	60.63	67.33	71.96	74.87	76.25	78.62	3.41 ± 0.04
0.3% PE	1.10	0.40	61.36	68.24	70.29	73.48	74.25	77.11	3.39 ± 0.08
0.7% PE	0.87	0.53	62.38	68.30	70.84	72.06	73.28	76.71	3.40 ± 0.06

* Transparency values have not any significant difference between films (*p* > 0.05).

**Table 3 foods-13-00538-t003:** Glass transition temperature (Tg), melting transition temperature (T_max_) and enthalpy of fish gelatin films without and with *Prosopis* extract (PE).

Film Type	Tg	T_onset_	T_max_	T_end_	∆H
Control	67.13	72.42	75.95	82.63	10.17
0.3% PE	59.78	50.79	63.47	71.02	10.94
0.7% PE	67.35	64.62	71.83	78.94	13.13

## Data Availability

Data is contained within the article.

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
