# Peer review of "Incorporation of Prosopis cineraria Extract Improved the Mechanical, Barrier and Antioxidant Properties but Not the Antibacterial Activity of Tigertooth croaker Fish Scale Gelatin Filmâ€"

_foods, 2024, doi:10.3390/foods13040538_

Round 1
Reviewer 1 Report
This article by Kord et al. deals with studying the effects of incorporation of prosopis extract on mechanical, barrier, antioxidant, and antibacterial properties of tiger tooth cracker fish scale gelatin film. Although the hypothesis of this study was investigated clearly and conclusions are drawn based on the results, the following points should be addressed to improve the quality of this manuscript:
1. The title should be modified as “Incorporation of Prosopis cineraria extract improved the mechanical, barrier and antioxidant properties but not the antibacterial activity of Tigertooth Croaker fish scale gelatin film.”
2. Key quantitative data on the studied properties should be included in the abstract.
3. Line 84, ‘produced of gelatin’ should be corrected as ‘produced with gelatin’.
4. Line 120-121, what is the volume of gelatin added?
5. All the equations should be numbered and eventually referred in the text with that respective equation number.
6. In section 2.6.4, the opacity is not abbreviated previously as “O” for its mention in the equation.
7. The subtopic 2.7.1 should be removed and instead subtopic 2.7 should be modified as “Microstructure analysis by scanning electron microscopy (SEM)”
8. The subtopic 2.8 should be modified as “Structural interactions by Fourier transform infrared spectroscopy-attenuated total reflectance (FTIR-ATR)”.
9. In subtopics 2.10 and 2.11, the title should be modified as “Determination of …” instead of “Measurement of …”
10. Line 232, ‘water vapor permeability’ should be ‘WVP’.
11. In Tables 2 and 3, the statistical significance of the data is missing.
12. The small font unimportant wavenumbers in Figure 2 should be removed and the axis labels should be made bigger to enable readability.
13. The key Tg values in Figure 4 should be marked in Figure 4.
14. The formatting of botanical/scientific names of natural products and microorganisms should be made in italics. Also, the superscript/subscript formatting is missing throughout the manuscript which needs to be corrected.
15. Few lines should be added at the end of conclusion to detail how the future work can be hypothesized based on this study’s outcomes.
Minor editing of English language required
Author Response
Dear respect reviewer
Thanks for you comments on our manuscript. We provide answers to your comments and questions as below:
- The title should be modified as “Incorporation of Prosopis cineraria extract improved the mechanical, barrier and antioxidant properties but not the antibacterial activity of Tigertooth Croaker fish scale gelatin film.”
Answer: This is done based on the comment.
- Key quantitative data on the studied properties should be included in the abstract.
Answer: The Key quantitative data added to the abstract.
- Line 84, ‘produced of gelatin’ should be corrected as ‘produced with gelatin’.
Answer: It is corrected.
- Line 120-121, what is the volume of gelatin added?
Answer: According to the requirement of 4% (w/v) solution, gelatin was used.
- All the equations should be numbered and eventually referred in the text with that respective equation number.
Answer: It is corrected.
- In section 2.6.4, the opacity is not abbreviated previously as “O” for its mention in the equation.
Answer: It is added to the text.
- The subtopic 2.7.1 should be removed and instead subtopic 2.7 should be modified as “Microstructure analysis by scanning electron microscopy (SEM)”
Answer: It is corrected.
- The subtopic 2.8 should be modified as “Structural interactions by Fourier transform infrared spectroscopy-attenuated total reflectance (FTIR-ATR)”.
Answer: It is corrected.
- In subtopics 2.10 and 2.11, the title should be modified as “Determination of …” instead of “Measurement of …”
Answer: These are corrected.
- Line 232, ‘water vapor permeability’ should be ‘WVP’.
Answer: These are corrected.
- In Tables 2 and 3, the statistical significance of the data is missing.
Answer: It is corrected in table 2 for Transparency values, but in table 3 it is not need based on the other reports about DSC analysis.
- The small font unimportant wave numbers in Figure 2 should be removed and the axis labels should be made bigger to enable readability.
Answer: We re-design the Figure 2, delete unimportant wave numbers from charts, and the axis labels made bigger for better understanding.
- The key Tg values in Figure 4 should be marked in Figure 4.
Answer: The Chart re-designed and the Tg values are marked in figure.
- The formatting of botanical/scientific names of natural products and microorganisms should be made in italics. Also, the superscript/subscript formatting is missing throughout the manuscript which needs to be corrected.
Answer: It is corrected.
- Few lines should be added at the end of conclusion to detail how the future work can be hypothesized based on this study’s outcomes.
Answer: We rewrite the conclusion and tried to highlight this comment.
Comments on the Quality of English Language: Minor editing of English language required
Answer: The whole manuscript read again and first correct by the authors, then give help from an English language editor for enhancing the English language and grammar writing of the article.
Thank you
Reviewer 2 Report
In this article submitted by Zeinab Kord et al., and entitled "Antioxidant and Antibacterial Properties of Tigertooth Croaker 2 Fish (Otolithes rubber) Scale Gelatin Film Incorporated with 3 Prosopis (Prosopis cineraria) Extract", the author endeavored to develop a novel edible film by utilizing scale gelatin films derived from Croaker fish and the Prosopis extract, as well as the antioxidant and antimicrobial activity of the film was also determined. Nevertheless, numerous issues and limitations have been identified with regards to this article.
1.What was the rationale behind the author's selection of 0.3% and 0.7% PE for film preparation? The data presented in this article supports the conclusion that a higher concentration of PE may offer greater advantages.
2.The author stated that Prosopis cineraria is a prolific source of phytochemicals, encompassing flavonoids, tannins, alkaloids, quinones, and phenolics. Consequently, it possesses commendable potential for exhibiting efficacious antioxidant activity. Additionally, there is limited information available regarding the antimicrobial properties of Prosopis cineraria. Why author consider to investigate the antibacterial activity of the film?
3. The "Results and discussion" section lack the information on the antimicrobial activity of the film.
4.In this article, the article includes images that are fuzzy, please substitute them with high-resolution ones.
The English writing in this article needs improvement.
Author Response
Dear Respect reviewer
We provide answers to your questions and comments in below:
1.What was the rationale behind the author's selection of 0.3% and 0.7% PE for film preparation? The data presented in this article supports the conclusion that a higher concentration of PE may offer greater advantages.
Answer: We first selected our desired concentrations according to the article by Wu et al. (Preparation, properties and antioxidant activity of an active film from silver carp (Hypophthalmichthys molitrix) skin gelatin incorporated with green tea extract) But after that, we tried other concentrations, but no significant difference was observed in the results; and that's why we continued with the same concentrations; Also, due to the flavor of the Prosopis plant, we were not able to increase the amount of its extract. Also, in the study of Sahraee et al (Effect of corn oil on physical, thermal, and antifungal properties of gelatin-based nanocomposite films containing nano chitin), almost similar concentrations were used. So, this is the reason of usage this amount of the extract in this study.
2.The author stated that Prosopis cineraria is a prolific source of phytochemicals, encompassing flavonoids, tannins, alkaloids, quinones, and phenolics. Consequently, it possesses commendable potential for exhibiting efficacious antioxidant activity. Additionally, there is limited information available regarding the antimicrobial properties of Prosopis cineraria. Why author consider to investigate the antibacterial activity of the film?
Answer: Considering that, in food storage, in addition to chemical reactions and antioxidant activities; reducing the activity of bacteria is an integral part of increasing the shelf life of food. It should focus on activities and reducing microbial activities in the food sector. Edible coatings can contain compounds that reduce the growth of bacteria and spoilage agents in food. So, the importance in this part will not be less important than the antioxidant. Beside, there have been many studies about this plant in different fields, such as Persia FA et al (Overview of Genus Prosopis Toxicity Reports and its Beneficial Biomedical Properties); Seturaman Prabha et al (Pharmacological potentials of phenolic compounds from Prosopis spp.-a review); Saad et al (Chemical constituents and biological activities of different solvent extracts of Prosopis farcta growing in Egypt); Nagalakshmi G. and Anuradha R (FT-IR ANALYSIS AND IN VITRO ANTIBACTERIAL ACTIVITY OF PROSOPIS JULIFLORA); Antibacterial properties have also been measured in most of the studies. so, it must assay the antibacterial properties of the films incorporated by the Prosopis sp extract.
- The "Results and discussion" section lack the information on the antimicrobial activity of the film.
Answer: It is added in the section 3.7
4.In this article, the article includes images that are fuzzy, please substitute them with high-resolution ones.
Answer: We substitute the SEM images by the high resolution one.
Comments on the Quality of English Language: The English writing in this article needs improvement.
Answer: The whole manuscript read again and first corrected by the authors, then give help from an English language editor for enhancing the English language and grammar writing of the article.
Thank you
Reviewer 3 Report
minor correction in language in terms of gramatical mistakes, phrases mistakes tec.
minor corrections
Author Response
Dear respect reviewer
The whole manuscript read again and first corrected by the authors, then give help from an English language editor for enhancing the English language and grammar writing of the article.
Thank you
Reviewer 4 Report
The Paper “Antioxidant and Antibacterial Properties of Tigertooth Croaker Fish (Otolithes rubber) Scale Gelatin Film Incorporated with Prosopis (Prosopis cineraria) Extract” reports an interesting experimental approach related to the effects of specific films. Furthermore, the authors noted that the incorporation of Prosopis extract into gelatin film showed an increase of mechanical and barrier properties and an increase of the in antioxidant activity.
The study is very interesting and original, considering the impact it could have at the industrial level, given that the plastic materials currently in use are harmful to both human health and the environment.
I suggest minor revisions.
1. All species names should be italicized: from title to throughout the paper. For example:P1 L10; P2L61, 75, 77, 84… The same for the names of the microorganisms (P5L210, 212…)
2. Abstract: P1L12. Change compara-tive in comparative
3. Introduction, P1L28. Change “Plastics” in “plastics”
4. Materials and methods: P2L93, change “labratory” in “laboratory”
I have a question about the preparation of prosopis extract. Authors should specify whether the leaves were taken from a single plant or from multiple plants. In all cases they should specify the reason for their choice. In my opinion, leaves from different plants should be chosen in order to exclude any problems (water stress, soil chemistry) that could affect the preparation of the extracts.
5. The authors should report figure 1 in a higher resolution.
6. Figure 2. In my opinion this image should be resized. Furthermore, the caption should contain the description of what is reported in the x and y axis.
7. Figure 3. The authors should better represent the histograms. The letters (A) and (B) should be placed on the outside of the graph. The letters on each bar should centered. The two graphs are represented with a different dimension and the written part is not centered, both in the x and in the y axis. Also in this case the caption should include more information relating to the statistical part as well.
8. The authors should expand the conclusions by highlighting what they add to the existing literature. This would allow to better highlight the final message of the whole paper.
Author Response
Dear respect reviewer
We work on manuscript corrections and provide answers to your questions and comments as below:
- All species names should be italicized: from title to throughout the paper. For example:P1 L10; P2L61, 75, 77, 84… The same for the names of the microorganisms (P5L210, 212…)
Answer: these are corrected in the whole body of the manuscript.
- Abstract: P1L12. Change comparative in comparative
Answer: It is corrected.
- Introduction, P1L28. Change “Plastics” in “plastics”
Answer: It is corrected.
- Materials and methods: P2L93, change “labratory” in “laboratory”
- Answer: It is corrected.
I have a question about the preparation of prosopis extract. Authors should specify whether the leaves were taken from a single plant or from multiple plants. In all cases they should specify the reason for their choice. In my opinion, leaves from different plants should be chosen in order to exclude any problems (water stress, soil chemistry) that could affect the preparation of the extracts.
Answer: We used several trees for sampling, mix of the leaves from different trees could help to normalize the bioactive compounds extract from the leaves and could be reproducible in the future studies.
- The authors should report figure 1 in a higher resolution.
Answer: We substitute the SEM images by the high resolution one.
- Figure 2. In my opinion this image should be resized. Furthermore, the caption should contain the description of what is reported in the x and y axis.
Answer: We re-design the Figure 2, Also the description of X and Y axis added inside the figure.
- Figure 3. The authors should better represent the histograms. The letters (A) and (B) should be placed on the outside of the graph. The letters on each bar should centered. The two graphs are represented with a different dimension and the written part is not centered, both in the x and in the y axis. Also in this case the caption should include more information relating to the statistical part as well.
Answer: The graph re-designed. The letters of (A) and (B) set the below of the graphs, the letters on each bar set centered, and more information added in the caption. The dimension of radical scavenging activity (%), and reduction power (landa) are different so these are cannot be represent same in both graphs.
- The authors should expand the conclusions by highlighting what they add to the existing literature. This would allow to better highlight the final message of the whole paper.
Answer: We rewrite the conclusion and tried to highlight this comment.